# A New Possibility for Fermentation Monitoring by Electrical Driven Sensing of Ultraviolet Light and Glucose

**DOI:** 10.3390/bios10080097

**Published:** 2020-08-12

**Authors:** Cleber A. Amorim, Kate C. Blanco, Ivani M. Costa, Estácio P. de Araújo, Adryelle do Nascimento Arantes, Jonas Contiero, Adenilson J. Chiquito

**Affiliations:** 1School of Sciences and Engineering, Av. Domingos da Costa Lopes, São Paulo State University (Unesp), 780 Jardim Itaipu, CEP 17602-496 Tupã, SP, Brazil; cleber.amorim@unesp.br; 2São Carlos Institute of Physics, University of São Paulo—Box 369, 13566-970, São Carlos, SP, Brazil; kateblanco@ifsc.usp.br; 3NanOLaB, Departamento de Física, Universidade Federal de São Carlos—UFSCar, Rodovia Washington Luiz, Km 235 Monjolinho, CP 676, CEP 13565-905 São Carlos, SP, Brazil; ivanimcosta@gmail.com (I.M.C.); estacio.paiva@gmail.com (E.P.d.A.); adryelle.fisica@gmail.com (A.d.N.A.); chiquito@df.ufscar.br (A.J.C.);; 4Institute of Chemistry, Araraquara. Rua Professor Francisco Degni, São Paulo State University (Unesp), Jardim Quitandinha, CEP 14800-060 Araraquara, SP, Brazil; 5Institute of Biosciences, Department of General and Applied Biology, São Paulo State University (Unesp), Rio Claro, Rio Claro, Av. 24-A, 1515 Bela Vista, CEP 13506-692 Rio Claro, SP, Brazil; 6Institute for Research in Bioenergy, São Paulo State University (Unesp) Rua 10, 2527 Santana, CEP 13500-230 Rio Claro, SP, Brazil

**Keywords:** glucose sensor, UV light sensor, nanowire biosensor, controlling fermentation

## Abstract

Industrial fermentation generates products through microbial growth associated with the consumption of substrates. The efficiency of industrial production of high commercial value microbial products such as ethanol from glucose (GLU) is dependent on bacterial contamination. Controlling the sugar conversion into products as well as the sterility of the fermentation process are objectives to be considered here by studying GLU and ultraviolet light (UV) sensors. In this work, we present two different approaches of SnO_2_ nanowires grown by the Vapor–Liquid–Solid (VLS) method. In the GLU sensor, we use SnO_2_ nanowires as active electrodes, while for the UV sensor, a nanowire film was built for detection. The results showed a wide range of GLU sensing and as well as a significant influence of UV in the electrical signal. The effect of a wide range of GLU concentrations on the responsiveness of the sensor through current–voltage based on SnO_2_ nanowire films under different concentration conditions ranging was verified from 1 to 1000 mmol. UV sensors show a typical amperometric response of SnO_2_ nanowires under the excitation of UV and GLU in ten cycles of 300 s with 1.0 V observing a stable and reliable amperometric response. GLU and UV sensors proved to have a promising potential for detection and to control the conversion of a substrate into a product by GLU control and decontamination by UV control in industrial fermentation systems.

## 1. Introduction

Large scale continuous industrial fermentation generates products of high commercial value through control variables process carried out in bioreactors. This control results in the generation of specific metabolic products with microbial growth, depending on the species or microbial lineage. Catabolism of complex carbohydrates by microorganisms generates monosaccharides used in glycolysis to produce energy for the cell, such as other fermentation products considering their conversion efficiency. The Monod model describes the microbial product yielded behavior converted from glucose (GLU) [1]; however, the industrial procedure requires real-time monitoring to optimize the process with the carbohydrate feed [2]. To achieve such purpose, ultraviolet radiation can be used as a tool to control the quality of this industrial microbial process by decontaminating the fermentation broth [3].

The continuous fermentation process also requires high levels of automation, and, for this, in addition to the well-developed control monitors, it is necessary to improve the systems for detecting different variables. The GLU sensor is frequently described, in literature, performing several roles in medical, biological, environmental and industrial sciences with clinical analysis: biological detection, environmental monitoring and food processing [4,5,6]. Furthermore, the GLU biosensors market for disease diagnosis system was estimated to be USD 13 billion in 2020 [7]. The conversion carbohydrate control in industrial fermentation processes is generally carried out by the microbial products detection and quantification, which are formed during the microbial metabolic process of multiplication. Non-enzymatic sensors have, when compared to conventional spectrophotometric sensors, considerable advantages, such as high sensitivity, easy instrumentation, low production cost, promising speed, among other features [8,9,10]. In addition, when these sensors are based on nanomaterials, GLU detection was ascribed to the catalytic properties of GLU oxidation in metallic nanostructures; however, the high cost and low stability still limit their practical applications. Therefore, developing a more practical and cheaper sensor for non-enzymatic GLU detection can optimize the industrial fermentation processes [11,12,13,14].

Variables such as microbial substrate and process cycle changes, which take place during industrial fermentation, may contain microbial contamination with high toxin loads. Consequently, it results in modifications and low quality of the substrate used, in addition to microbial inhibition and low quality of the product formed. Ultraviolet light wavelengths ranged between 200 and 300 nm (Ultraviolet - UV-C) can provoke structural distortion of cellular deoxyribonucleic acid (DNA), which leads to its death due to identification errors during genetic sequencing in bacterial multiplication. Its efficiency was evidenced many times for pathogenic microorganisms, such as *Salmonella typhimurium, Escherichia coli* and *Salmonella* sp. Decontamination technology processes involving UV-C have been extensively studied and often described for fresh food due to their broad-spectrum antimicrobial advantages. In addition, cost-benefit, ease of use and environmental protection turn them into potential alternative chemical and thermal disinfection methods [15]. Based on this context, the decontamination by these methods becomes fundamental to improve the performance of the process [16]. However, UV light exposure is related to a low penetration effect on surfaces and deterioration in product quality when under a long exposure period. For instance, prolonged exposure of freshly cut apples in UV-C led to cell surface destruction due to their dehydration, causing changes in the fruit flavor. Therefore, considering that UV effectiveness against microorganisms depends on factors such as wavelength, dose, properties of exposed substances and cellular structures, it is essential to precisely control intrinsic factors to the technique to achieve desired death without altering products of interest in microbial fermentation systems [16,17,18]. In general, a higher lethal efficacy is achieved at a closer wavelength of DNA absorption or higher doses applied [16].

Although the monitoring test descriptions for GLU and UV light can be found in detail in literature [14,19,20,21,22,23], procedures should be modified due to each particular characteristic of process. Thus, alternative methods for monitoring these processes have great appeal, both academic and commercial. Conductive type electrical sensors, i.e., those that change the conductivity of the active layer, showed a significant advantage in detecting microbial metabolism. Metal oxide-based devices have been known since the middle of the last century, when the reaction of metal oxides with the surrounding atmosphere was observed [24,25,26,27]. Metal oxides can be n-type (the majority of carriers are electrons) and p-type (the majority is the ”holes”). Among the metal-oxide-semiconductor (MOS), tin dioxide (SnO_2_) has been explored as a sensitive material to different stimuli. SnO_2_ is classified as an n-type semiconductor presenting a 3.6 eV wide-bandgap at an environment temperature that enables its application in a variety of technologies. The sensing principle of SnO_2_ is based on changes in the electrical conductivity influenced by adsorbed oxygenated species on its surface [28,29].

Nanostructures based on tin oxide show high performance associated with low dimensionality and its morphology. Moreover, nanostructured metal oxides can be applied in both their pure (intrinsic) and doped forms raising their potential as chemical and biological sensors. Besides, they have distinct characteristics, such as high surface-to-volume ratio, modified-surface work function, high surface reactivity, high electron mobility and high effectivity in retaining biomolecules [8,30,31,32]. In addition, nanowires can be incorporated relatively easily into microelectronic devices. It is known that the transport properties of nanostructures (transparent conductive oxides) depend heavily on the surface of these materials. The nanowire surface is responsible for the process of adsorption and desorption of oxygen and glucose molecules that results in the capture of electrons, thus changing the electrical conductivity of the device. The electrical-guided fermentation using biosensors in bioreactors can provide the possibility of cultivation monitoring in a system with integrated sensors in fermenter cover ports and if possible, contain a protection system to prevent mechanical impurities from causing a loss in sensitivity. The sensors hypothesis can be implemented in bioreactor scales and enable the specific monitoring of microbial growth and metabolite products in fed-batch cultures.

We present an approach that modifies SnO_2_-based device geometry to monitor both UV, widely used for microbial decontamination processes, and GLU consumption in continuous fermentation processes.

## 2. Materials and Methods

The nanowires were grown according to the experimental method described by Costa et al. according to the Vapor-Solid-Liquid method [33,34]. Two types of devices were built: (*i*) active electrodes based on SNO_2_ nanowires—GLU sensor; (*ii*) multiple SnO_2_ nanowires—UV light sensor. The device (*i*) was constructed following the conventional photolithography procedure [23,29,35,36]. Initially, metal electrodes (separated by 5 μm) were defined on a Si/SiO_2_ (500 nm thick oxide layer) substrate. Then a small region (10 μm in diameter) was selected. Catalyst nanoparticles were formed by evaporating a thin layer of gold (2 nm), and they were used as the preferred site for vapor precursor elements adsorption. The precursor tin metal powder (Aldrich, purity >99.99%) was placed in an alumina crucible and placed in the center of a tubular reactor (Lindberg Blue-M). The substrates with the electrodes and a small region covered with gold were placed 8 cm downstream from the precursor powder. The synthesis temperature was adjusted to 950 °C (heating rate 20 °C min^−1^) and remained at this value for 90 min. The pressure inside the tube was controlled by a vacuum pump, and a mixture of oxygen/argon with a constant flow of 100 sccm was used to transport the precursor vapor to the synthesis region. After this process, the oven was naturally cooled to room temperature. The device (*ii*), based on multiple SnO_2_ nanowires, was manufactured by dispersing the nanowires on thermally oxidized Si wafer with an oxide layer (SiO_2_) of 500 nm thickness where interdigitated electrodes were previously microfabricated using lithographic techniques (Au/Ni-In, 70−30 nm). The final device was inserted into an annealing tube furnace filled with inert argon atmosphere at 450 °C for 10 min.

The experimental setup for the characterization of the glucose sensor is shown in Figure 1a: it is constituted of three electrodes: (i) working electrode formed by a set of nanowires grown by the VLS method; (ii) reference electrode and (iii) common electrode; the last two were made of a 100 nm layer of titanium. Electrical characterization was performed by a four-probe measurement, whose scheme is shown in Figure 1b. The electric current–voltage characterization was performed by applying an electric voltage (−0.25–1.00 V) between the working electrodes and the common electrode; the electrical current between the working electrode and the reference electrode was then measured. Measurements regarding the solvent and glucose solution were performed by inserting 20 μL of solution into the device, closing the electrical circuit. The glucose solution was tested in concentrations of 1 to 1000 mmol. In addition, current–time characterization was performed, maintaining a fixed voltage of 1.0 V and a glucose solution of 10 mmol. The UV light sensor based on an interdigitated structure with the SnO_2_ nanowire “film” is depicted in Figure 1c. Tin oxide nanowires photoresponse as UV light sensor was obtained using a SMU unit (Keithley 2400) and a light chopper to generate the on and off states. As a UV light source, a Cole-Parmer UVGL-15 lamp −2 mW/cm^2^, with main spectral lines at 254 and 365 nm was used [37].

## 3. Results

Figure 2 depicts the structural data and chemical composition of the as-grown SnO_2_ nanowires. As first examination of the samples, SEM images were taken. Figure 2a reveals nanowires with an excellent uniformity on diameters and lengths. Figure 2b shows the results for the bulk material crystal structure by X-ray diffraction (XRD) and Miller indices are indicated on each diffraction peak. The result points out a good crystalline quality in close agreement to the JCPDS Card No. 41-1445 (represented by blue circles), indicating that the as-grown material presents a tetragonal SnO_2_ structure (space group P42/mnm) (Figure 2c). Figure 2 presents the corresponding Energy dispersive X-ray spectroscopy (EDX) spectrum. It ratifies the chemical composition of a nanowire, indicating the presence of Sn and O (Si signal was generated by a substrate).

Figure 3a displays the GLU biosensor based on SnO_2_ nanowire current–voltage curves in different conditions: (i) only solvent (DI water) and (ii) different GLU concentrations −1, 10, 100 and 1000 mmol. This verified the effect of an extensive range of GLU concentration on the sensor responsivity. However, for concentrations of 100 and 1000 mmol, the increase in the electric current was relatively high. This effect shows the high sensing character of devices for the presence of the analyte (GLU).

Figure 3b presents the sensibility of GLU sensor based on the SnO_2_ nanowire film. Parameters such as the sensitivity (in our case, obtained for 1.0 V applied voltage (*Vap*)), can be defined in terms of conductivity or the electric current, and written as
(1)$$S_{g}=\frac{G_{NW}-G_{an}}{G_{an}}=\frac{I_{NW}-I_{an}}{I_{an}}$$
where *G_NW_* and *G_an_* are the electrical conductance of biosensor based on the SnO_2_ nanowire and GLU solution, and *I_NW_* and *I_an_* are the corresponding currents, respectively. It can be seen in Figure 3b that the conductivity of the device considerably improves with increasing GLU concentration.

The enhancement was found to have a gain factor of up to 25,000 for the SnO_2_ sensor to 1000 mmol glucose concentration. It can be justified by glucose molecular adsorption on the surface of the SnO_2_ nanowire by oxygen interactions, which results in the increase in the electrical conductivity. Accordingly, it is straightforward to note that increasing the GLU concentration, the electrical current increases. In this study, the biosensor based on the SnO_2_ nanowire allowed a remarkable rise in the surface area subjected to the aqueous solution containing glucose. The electrical current variation depends on how much electron activity is affected by the amount of chemisorbed oxygen species from the carbohydrate [38]. In this way, the hydroxyl groups present in the GLU molecule interact with the SnO_2_ nanowire film surface, causing notable changes in the device resistivity and electric current.

During the fermentation process, the growth of biomass, consumption of substrate and generation of products arise. Mathematical models have shown that bacterial growth is dependent on the energy generated during the nutrient conversion [38]. Glucose is converted from polysaccharides with distinct yields, considering factors such as microorganisms, processes and desired products. Microorganisms can totally consume GLU in a sigmoidal curve similar to the growth of biomass or decrease when glucose is depleted; however, this will depend on the type of fermentation: continuous or discontinuous. In this study, we demonstrated the possibility of detecting the amount of 1–1000 mmol of glucose which is correlated with the formed product rate [39].

Figure 4 exhibits the response (sensitivity) of nanowire devices based on SnO_2_ for the detection of UV light and GLU sensor, response and recovery time analysis, respectively. The same sensor performing the UV light detection and afterwards glucose analysis was used.

Figure 4a,b show a typical amperometric response curve for sensors based on SnO_2_ nanowires under UV light excitation and glucose, respectively. Ten cycles of 200 s were performed (100 s UV light illumination and 100 s for device recovery) with *V_ap_* = 1.0 V (Figure 4a). Regarding the Glucose sensor, ten cycles of duration 200 and 80 s in the presence of 1000 mM glucose and 120 without the presence of the analyte were carried out. The applied voltage in this experiment was also *V_ap_* = 1.0 V (Figure 4b). It was observed, for both cases, a well-defined, stable and reproducible amperometric response. The GLU sensor calibration curve is shown in Figure 4b. The photocurrent, when sensors were under UV irradiation, was the due to electron excitation through the SnO_2_ band gap (3.68 eV) [40,41]. The transport properties, regarding nanostructured-based transparent conductive oxides, strongly depended on the oxygen vacancies action. In the first case, under the dark condition, oxygen molecules were adsorbed on the surface of the nanowire, capturing electrons according to the following equation:(2)$$O_{2\left(gas\right)}+e^{-}\;\to \;O_{\left(ads\right)}^{-2}.$$

The electron capture gave rise to a non-conductive depletion layer under the surface. When the nanowire device was under UV light, electrons were promoted to the conduction band and holes were generated in the valence band. These holes migrated towards the surface and recombined with the electrons trapped on the surface. After illumination, some of those photo-generated carriers tended to move towards the nanowire surface to discharge and absorb the oxygen ion.
(3)$$O_{\left(ads\right)}^{-2}+h^{+}\;\to \;O_{\left(gas\right)}^{2}$$

When UV light illumination ceased, the environment oxygen molecules were adsorbed and electrons were trapped, increasing the surface potential barrier once more given that, after a certain time interval, the electric current returned to its initial value. Regarding the decoupling of e–h pairs, an inner conductive channel was formed within the nanowire, and the applied external voltage separated the e–h pairs, resulting in a considerable increase in the electric current.

Essential parameters of sensory analysis are related to the response and recovery times. The first is associated with the stabilization of the sensitivity after the injection of the analyte to be studied. The latter is regarding the time at which the resistance returns near its original value [42,43]. The analysis of the response and recovery times was performed using two methodologies: (i) qualitative, referring to percentages of maximum and minimum estimated values and (ii) by adjustment of theoretical models. It was established that the response time was the one where the sensitivity reached 90 % of the maximum value, whereas two recovery time values should be taken for both devices: at 10% and 5% of the initial response value, in dark conditions. Figure 4c,d show these results for both sensors, UV and glucose, respectively. The UV light sensing response and recovery time curves at 1.0 V applied are displayed in Figure 4c. All curves were analyzed, and the response and recovery times were obtained by the experimental data fitting according to the following expressions:(4)$$S\left(t\right)=\frac{\Delta R}{R}=S_{0}\left(1-e^{\frac{-t}{\tau_{on}}}\right)$$
(5)$$S\left(t\right)=\frac{\Delta R}{R}=S_{0}+Ae^{\frac{-t}{\tau_{off-1}}}+Be^{\frac{-t}{\tau_{off-2}}}$$
in which *S*_0_ is the maximum sensibility, *t* is the time, *A* and *B* are constants. Figure 4c presents the UV light sensing sensibility variation subjected to a 180 s pulse at a V_ap_ = 1.0 V. One can observe that the response time was 6.3 ± 3.2 s and the recovery times were 2.3 ± 0.5 and 44.8 ± 3.0 s, to 10% and 5% of maximum value, respectively. Both methods, qualitative and fitting showed similar values within the same order of magnitude. The presence of two recovery times for the UV light sensor was analyzed according to Equation (4), in which *t*_off-2-U.V._ was related to a rapid change in the concentration of carriers when the incidence of light is ceased. Time *t*_off-2-U.V._ was related to trapping and releasing these carriers due to oxygen vacancies and other energy levels within the bandgap.

Figure 4d depicts the result obtained for the GLU sensing sensitivity response as a function of time. This result presented at a higher recovery time when compared to the UV sensor; the same behavior was observed for the recovery time. Both *t*_on-Glucose_ response time and *t*_off-Glucose_ recovery time showed agreement between the two methodologies. The sensor response (sensibility) depends on how much electron activity is modified by the amount of chemisorbed oxygen species from the carbohydrate. Therefore, the hydroxyl groups present in the GLU molecule interact with the SnO_2_ nanowire film surface, provoking notable changes in the resistivity device and electric current. Table 1 summarizes all times obtained by the two methodologies presented to both UV and glucose sensors.

Figure 5 displays a schematic of the three different conditions the devices were exposed to: the first one (Figure 5a) is the dark condition, in which O^2−^ molecules, represented by red circles, were adsorbed on the SnO_2_ surface. When under UV illumination, electron–hole pairs were formed leading to a combination of the holes with the O^2−^ molecules of the surface and its release as O_2_ atoms, displayed in Figure 5b. Figure 5c represents the third condition, in which the surface film formed an Electric Double Layer (EDL) with the solution containing GLU. The studied solution contained free ions, such as H^+^, that interacted with the adsorbed O^2−^ atoms in the surface, leading to their dessortion. As a consequence, the desorption enabled −OH interaction, present in the GLU molecule, with the film surface. All conditions provoked changes in the conductivity of the devices [44].

The industrial fermentation efficiency of ethanol from sugar is dependent on bacterial contamination, which is often excessive. Sterilization processes on industrial scales are complex and expensive to achieve in operations that generally take a long time. Sensors, in which the presence of sugar and decontaminating light are considered, can generate an operation improvement due to the control of these factors, decisive for the final quality of the process [45]. For example, in the scope of what Fatima et al. introduced as an alternative for the cost reduction in the method [46]. The modification of components, material and time of manufacture can be an option for cost reduction in the nanowire biosensor. It is known that traditional UV detection methods are based on the use of fiber optic sensors, and for the consumption of sugar, it is detected by indirect methods such as the production of biomass by optical spectroscopy, which are methods of higher costs than those described for the proposed biosensor in this study. Previously, the performance of the GLU and UV sensors were observed independently. Figure 6 presents a response graph for the device based on SnO_2_ nanowires in different configurations: only the SnO_2_ nanowires; under UV light; the combined effect of the UV light with 1 mol of GLU.

For both UV light and GLU, the transduction process occurred through the release of electrons, changing the conductive material conductivity. However, the sensor exposure to the combined effect of GLU and UV light led to a lower electrical signal when compared to the isolated effect of glucose. One explanation may be related to the glucose breakdown. A study was performed by Cavicchiol et al. [47], in which irradiated organic samples by UV light were presented as useful in the decomposition and derivatization of analytes. For instance, GLU photo-decomposition generated as a CO_2_ product was reported by Kaneko et al. [48] and Luigi Da Vià et al. [49], which contributes to the electrical response decrease in metal oxides, as was demonstrated by Yadav et al. [50].

## 4. Conclusions

The biosensors studied here showed potential for detecting analytical routes of conversion of GLU into biomass and microbial products studied in fermentations, which showed high sensitivity of the primary source of nutrients explored in these systems. The biosensors analyze, in addition to converting a substrate into a process product, possibilities to detect these changes in the maintenance of the decontamination by ultraviolet light activated with the consumption of all GLU, after which the process must be started with the insertion of GLU, and microbial strain specifies the fermentation studied. These GLU and UV biosensors were able to be applied in fermentation processes in precise and economical ways to monitor microbial conversion in products considering the biochemical parameters of these processes, helping their synergistic efficiency with their decontamination.

## Figures and Tables

**Figure 1 biosensors-10-00097-f001:**
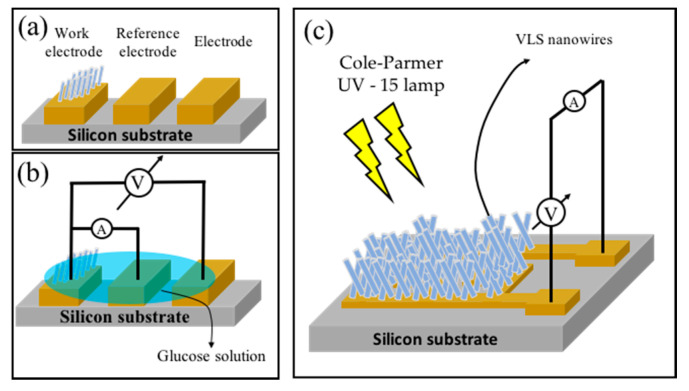
(**a**) Sketch of the used devices and the electrical connections and (**b**) diagram of the 4 probe setup which was performed to perform the biosensor electrical measurements and (**c**) setting the UV light sensor.

**Figure 2 biosensors-10-00097-f002:**
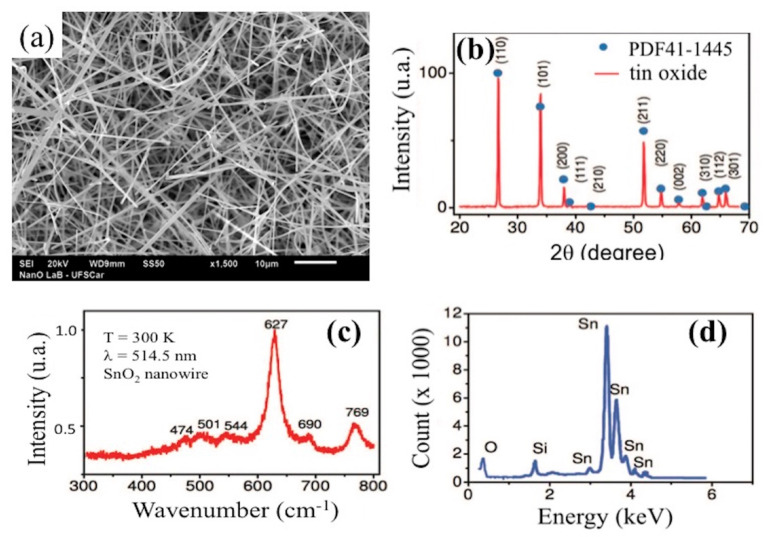
Structural data and chemical composition of the SnO_2_ nanowires: (**a**) SEM image of as-grown nanowires exhibiting lengths of tens of micrometers and (**b**) XRD pattern obtained agreeing with JCPDS Card No. 41-1445; (**c**) EDX analysis confirmed that the nanowires are composed only of Sn and O, and (**d**) room-temperature Raman spectra of SnO_2_ nanowires.

**Figure 3 biosensors-10-00097-f003:**
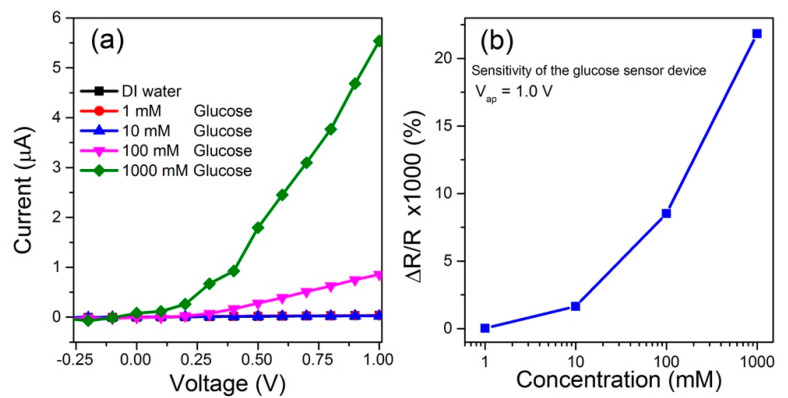
Experimental results concerning the biosensors electrical characterization: (**a**) current–voltage characterization for different glucose (GLU) concentrations for biosensor based on a SnO_2_ nanowire; (**b**) sensitivity vs. GLU concentration for biosensor under V_ap_ = 1.0 V.

**Figure 4 biosensors-10-00097-f004:**
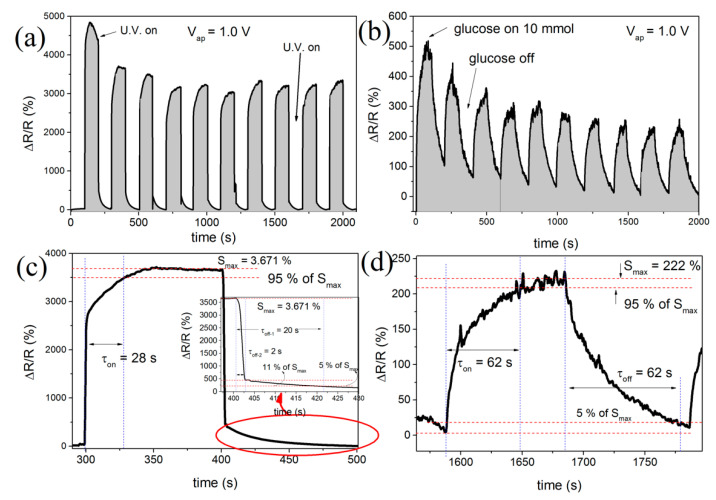
Sensors results to (**a**,**b**) sensibility UV, photoresponse characteristics and GLU sensor based on SnO_2_ nanowire, respectively. In (**c**,**d**) the response time graphs for both UV and GLU sensors are shown.

**Figure 5 biosensors-10-00097-f005:**
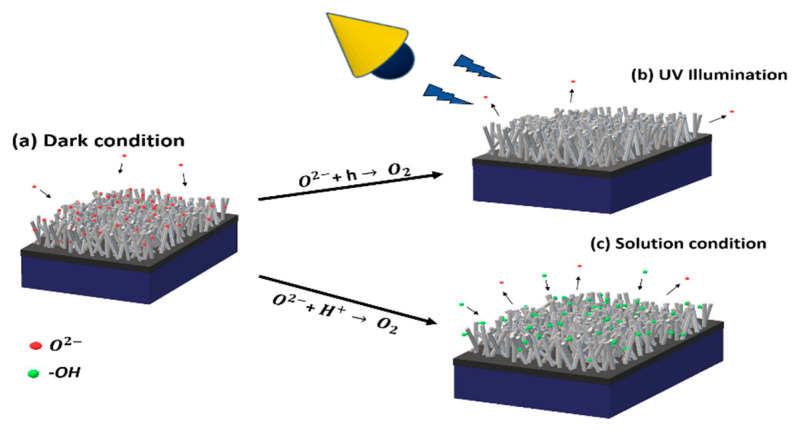
The device under three different conditions: (**a**) in the dark, in which O^2−^ atoms (red circles) were adsorbed on the surface of the NW film; (**b**) under UV illumination where O_2_ atoms were desorbed; (**c**) the device covered in a solution containing GLU, where O_2_ atoms were desorbed and –OH atoms (green circles) then interacted with the surface film. Arrows indicate which chemical–physical reaction took place in the process.

**Figure 6 biosensors-10-00097-f006:**
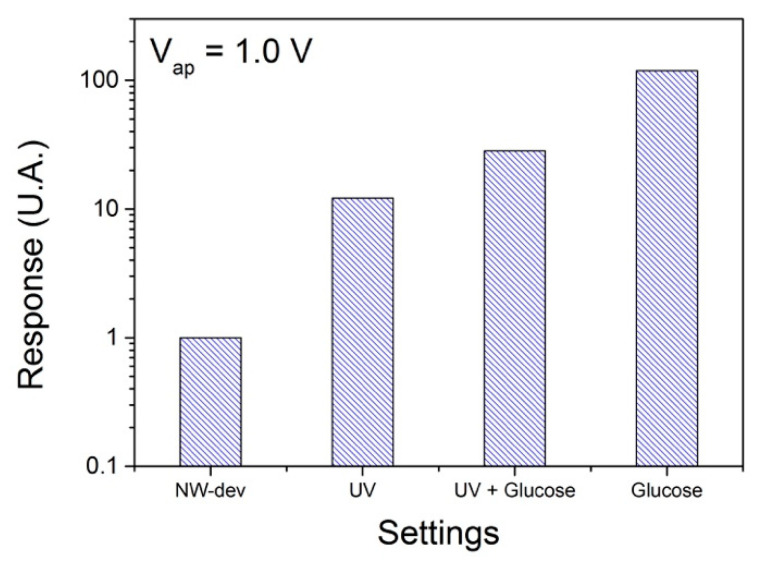
Response bar-graph of device based SnO_2_ nanowire sensor in four different settings: only nanowire (NW-dev); under UV (UV); combined effect between UV light and 1 mole of GLU (UV + Glucose) and under 1 mole of GLU (Glucose).

**Table 1 biosensors-10-00097-t001:** Table with response time and recovery times value to UV and GLU sensor.

UV Light Sensor				Glucose Sensor	
Recovery Times (s)				Response Time (s)	
	τ_on-UV_	τ_off-1-UV_	τ_off-2-UV_	τ_on-Glucose_	τ_off-Glucose_
By fitting	1.23 ± 0.61	0.48 ± 0.20	28.55 ± 2.98	19.36 ± 6.71	38.48 ± 4.87
Qualitative	6.3 ± 3.2	2.3 ± 0.5	44.8 ± 3.0	32.4 ± 4.6	43.8 ± 8.6

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
