# Peer review of "A New Possibility for Fermentation Monitoring by Electrical Driven Sensing of Ultraviolet Light and Glucose"

_biosensors, 2020, doi:10.3390/bios10080097_

Round 1
Reviewer 1 Report
The objective of this atticle is controlling the sugar conversion into products as well as the sterility of the fermentation process. But the manuscript discusses only the possibility of two different approaches of SnO2 nanowires grown in industrial fermentation. Do the character of GLU sensor and UV sensor meet the expected requirements?
Acronyms should be given full names when they first appear,such as VLS in the abstract.
Fifure 3 shows that the electric current increases with the increase of voltage and the sensitivity increases with the increase of concentration. So, how to suppres the influence of concenteation measurement of the voltage?
There are no results of repeatability and sensitive limit.
Author Response
Please check the cover-letter

Reviewer 2 Report
The authors propose using a sensor based on nanostructures to monitor the efficiency of the industrial fermentation process.
The manuscript is carefully prepared, the results are well described.
However, it is not clear from the introduction why the use of nanostructures is preferred. What are the advantures?
It is also not described how sensors can be integrated into a system that is used for industrial fermentation. Usually these are large containers. Perhaps the sensors are supposed to be used for analysis of sampling from bioreactor?
The authors also use glucose solution in their experiment. But if the sample is liquid from a fermentation vessel, then this sample contains mechanical impurities (for example, fiber) that can be non-specifically adsorbed onto the surface of nanowires and the sensor will lose its sensitivity.
Authors should more clearly describe how the results of their research can be applied in the industrial process.
Author Response
Please check the cover-letter

Reviewer 3 Report
Summary: Amorim et al. has presented an economic yet effective approach that modifies SnO2-based device geometry to monitor both Ultraviolet (UV) based decontamination and Glucose (GLU) consumption in continuous fermentation process.
Following are the comments which should be addressed for further improvement of the overall quality of the manuscript.
(1) In Abstract, full-form of VLS method is required
(2) Page 2, line 13, “product’s detection and quantification” – which product is referred here. Please clarify.
(3) Introduction of enzymatic and non-enzymatic sensor is required. How these sensors are related to either GLU or UV sensors.
(4) Please specify the term “limiting detection”. What features are limited in this case?
(5) The statement “However, enzymatic and non-enzymatic sensors present high-cost” is contradictory to low production cost as mentioned later.
(6) Authors stated that a more functional and cheaper sensor is essential for industrial fermentation process. Please elaborate what functional is referring to.
(7) Please specify how the proposed approach is a low-cost alternative in the discussion.
(8) Relevant references are required in the first paragraph of the Introduction section. Following are few examples:
Levenspiel, Octave. Biotechnology and bioengineering 22, no. 8 (1980).
Vojinović, V., J. M. S. Cabral, and L. P. Fonseca. Sensors and Actuators B: Chemical (2006)
Gouma, M., E. Gayán, J. Raso, S. Condón, and I. Álvarez. BioMed research international (2015)
(9) Several typos and grammatical errors. I would recommend to rephrase the sentences where authors have used ‘s or s’.
(10) Please see “Review of cost reduction methods in photoacoustic computed tomography” by Afreen Fatima, Photoacoustics and explain the benefit of the proposed method compared to the photoacoustic technology.
Author Response
Please check the cover-letter

Reviewer 4 Report
The present work reported by Amorim et al. is well written with supported findings and it is a good addition to the current literature. The results reported in the sensing part for GLU and ultraviolet light (UV) is convincing and presented with good explanation for the mechanism. However, the authors may consider to move characterization of nanowires and the device fabrication schematic under results section.
Author Response
Please check the cover-letter

Round 2
Reviewer 2 Report
The authors responced to all comments